# Strong Synergic Growth Inhibition and Death Induction of Cancer Cells by *Astragalus membranaceus* and *Vaccaria hispanica* Extract

**DOI:** 10.3390/cancers14235833

**Published:** 2022-11-26

**Authors:** Zoya Cohen, Yair Maimon, Noah Samuels, Hadar Brand, Aaron Sulkes, Baruch Brenner, Raanan Berger

**Affiliations:** 1Felsenstein Medical Research Center, Beilinson Hospital, Rabin Medical Center, Petach Tiqva 49100, Israel; 2Tal Center for Integrative Medicine, Institute of Oncology, Sheba Medical Center, Ramat Gan 52621, Israel; 3Center for Integrative Complementary Medicine, Shaare Zedek Medical Center, Faculty of Medicine, Hebrew University of Jerusalem, Jerusalem 9103102, Israel; 4Institute of Oncology, Davidoff Cancer Center, Beilinson Hospital, Rabin Medical Center, Petach Tiqva 49100, Israel; 5Sackler Faculty of Medicine, Tel-Aviv University, Ramat Aviv 69978, Israel; 6Institute of Oncology, Sheba Medical Center, Ramat Gan 52621, Israel

**Keywords:** anti-cancer, synergic drugs, botanical extracts, drug screening, cell cycle

## Abstract

**Simple Summary:**

Many anti-cancer drugs were developed from botanicals used in traditional medicine. However, modern anti-cancer therapy is often based on drug combinations that provide synergic anti-cancer effects. The aim of our study was to use costumed activity-based screening of traditionally used botanicals, which were previously screened by us for anti-cancer effects, for synergic anti-cancer combinations. We found a combination of two botanicals with very strong synergic effect on cancer cells. This effect was observed in all tested human cancer cell lines and in primary tumor cells. Our results suggest that the treatment interferes with the cell cycle of cancer cells, causing them to accumulate in the G2 phase of the cell cycle and leading to apoptosis. Further research of this combination may lead to the development of new effective anti-cancer therapies.

**Abstract:**

We present here a new, classification-based screening method for anti-cancer botanical combinations. Using this method, we discovered that the combination of *Astragalus membranaceus* and *Vaccaria hispanica* (AV) has strong synergic anti-proliferative and killing effects on cancer cells. We showed that AV induces the hyper activation of proliferation and survival pathways (Akt and ERK1/2) and strongly downregulates the cell cycle control proteins p21 and p27. Moreover, FACS analyses revealed that AV induces accumulation of cells in G2/M phase, supported by accumulation of cyclin A. Taken together, our results suggest that AV interferes with the cell cycle in cancer cells, leading to accumulation in G2/M and apoptosis. Further studies are needed to validate the generalizability of the anti-cancer effect of the AV combination, to fully understand its mechanism of action and to evaluate its potential as a new anti-cancer treatment.

## 1. Introduction

The botanical world is a natural stock of bioactive materials, providing modern medicine with an enormous amount of potentially useful phytochemicals, some of which turned over the years into conventional medications [1]. Among other areas, cancer treatment has been greatly advanced by the development of cytotoxic drugs such as camptothecin derivatives (topotecan, irinotecan), taxanes (paclitaxel, docetaxel), vinca alkaloids (vinblastine, vincristine) and epipodophyllotoxin derivatives (etoposide, teniposide), originating from botanical sources [2]. Discovery of new botanical-derived anticancer drugs is an intense ongoing process and several agents such as flavopiridol, roscovitine, betulinic acid and silvestrol are currently in clinical or preclinical development [3]. Thus, screening the medicinal botanicals for anti-cancer effect is important not only for evidence-based proof of efficacy of integrative and alternative medicines, but for new conventional drug discovery as well.

Astragalus membranaceus, known as “Huang Qi” in Chinese traditional medicine, is one of the most intensively studied traditional botanicals. Its major constituents are flavonoids, alkaloids, saponins and Astragalus polysaccharide (APS), composed of heterogeneous carbohydrates [4]. APS possesses multiple pharmacological actions, and has a great potential for further drug development due to its low toxicity [4]. Its functions include anti-aging and life span extension, which can, at least in part, be explained by APS’s antioxidant abilities; regulation of lipids levels in blood; numerous antiviral and antibacterial effects; and anti-fibrosis effects in several organs, partially by inhibiting collagen production and down regulating TGF-β1 [4,5,6,7]. APS regulates blood glucose levels and partially protects mice from type I diabetes mellitus onset, interfering with β-cells apoptosis and regulating the ratio of T helper cells [4,8]. APS has numerous positive effects on type II diabetes mellitus, including attenuation of high expression of PTP1B, a negative regulator of insulin signaling, induced by ER stress, reduction in blood glucose levels and protection of islet β cells [4,9]. The multiple effects of APS on the immune system include improvement of immune organ index and increasing their weight, balancing cytokines production under different conditions and stimulation of IgA, IgM and IgG expression. APS facilitates maturation, growth and antigen-presenting ability of dendritic cells, the key activators of immune response, increases the proliferation and the maturation of B and T lymphocytes and balances the ratio of T cells subgroups [4,10,11]. APS was also shown to enhance humoral response to immunization by reducing INFγ^+^ natural killer (NK) and natural killer T (NKT) cells populations, and increasing IL-4^+^ NK population [12]. Besides immunity enhancement, anticancer properties of APS include direct inhibition of cancer cells proliferation in vitro and in vivo and apoptosis induction. APS can also attenuate the metastasis of cervical cancer C33A cells and the migration and invasion of cervical cancer C-41 cells by inducing a reduction in matrix metalloproteinase 2 levels; metastasis of hepatocellular carcinoma H22 cells is inhibited by APS by down regulation of NOTCH1, while mouse Lewis lung cancer cells metastasis is attenuated through NF-κB and MAPK pathways inhibition [4,13,14,15,16,17]. Finally, APS may have a role in reducing the toxicity effects of chemo-and radiotherapy [4]. Further studies on precise mechanisms of the above effects and detection of individual bioactive components of APS can greatly promote the integration of traditional Chinese medicine into clinical practice.

Vaccaria seeds, known as “Wang Bu Liu Xing”, are traditionally used in Chinese medicine for treating amenorrhea, breast and urinary tract infections and to stop bleeding. Their active compounds include triterpene saponins, small cyclopeptides, flavonoids and crude polysaccharide [18]. Crude polysaccharide from *Vaccaria* seeds was previously shown to inhibit benign prostatic hyperplasia in mice model by reducing the prostatic and the testicular indexes [19]. Two compounds from the aqueous extract, vaccarin and apigenin-6-C-arabinosyl glucoside, were found in prostate tissue of a BPH rat model, suggesting their involvement in BPH inhibition [18]. Several compounds were also shown to exhibit anticancer activity in vitro [20].

Over the years, single drug therapy has gradually cleared its way to complex treatment protocols, affecting several targets simultaneously [21]. This strategy helps to overcome cancer cell survival through development of drug resistance and often has a synergic anti-cancer effect. However, finding synergic drug combinations usually requires deep understanding of the molecular mechanism of action of each drug, and thus can be applied only to intensely studied medications. Moreover, combinations of non-toxic or mildly toxic substances are left outside the focus, with major effort being made to test combinations of mostly effective drugs. Finally, medicinal botanicals are usually used in formulas, combined from several herbs according to traditional medicine rules, and research frequently focuses either on single botanicals or on a whole formula [22,23,24]. Thus, non-canonical two-herb combinations usually receive less attention.

In the current study, we employed a new strategy to find potentially useful combinations of medicinal botanicals. Previously we have shown that the majority of plants with an in vitro anti-cancer activity cause cancer cell death through induction of reactive oxygen species (ROS) [25]. We further showed that botanicals with ROS-independent anti-proliferative activity display different sensitivity profiles on a panel of human cancer cell lines, suggesting distinct mechanisms, as opposed to ROS-inducing botanicals, which display similar sensitivity profiles [25]. Classification of botanicals according to their anti-cancer effect and its dependence on ROS allowed us to perform costumed screening, taking into account the specificity of action of each botanical. Here, we present our findings on a potentially useful combination revealed by this screening and describe our initial findings on the possible mechanism of action.

## 2. Materials and Methods

### 2.1. Antibodies and Reagents

Primary antibodies: mouse anti-PARP-1, rabbit anti-phospho-Ser473 Akt, rabbit anti-phospho-Thr308 Akt, rabbit anti-Akt, rabbit anti-phospho-ERK1/2, rabbit anti-ERK1/2, rabbit anti-phospho-MEK1/2, mouse anti-MEK1/2, mouse anti-p21waf1, rabbit anti-p27, were from Cell Signaling Technologies (Boston, MA, USA). Mouse anti-human GAPDH and rabbit anti-cyclin A were from Santa Cruz Biotechnology (Dallas, TX, USA). Secondary antibodies: horseradish peroxidase (HRP)—conjugated goat anti-rabbit and goat anti-mouse IgG (H + L) antibodies were from Jackson (Baltimore Pike West Grove, PA, USA). EZ-ECL enhanced chemiluminescence detection kit, RPMI1640 medium, L-glutamine, fetal bovine serum, trypsin, antibiotics and phosphate buffered saline (PBS) were from Biological Industries (Beit-Ha-Emek, Israel). Propidium iodide, phosphatase inhibitor cocktails 2 and 3, sulphorodamine B (SRB), trichloroacetic acid and acetic acid were from Sigma Aldrich (St. Louis, MO, USA). ZSTK474, Selumetinib and AEW-541 were from Selleckchem (Houston, TX, USA). Complete mini protease inhibitor cocktail, RNAse A and Triton X-100 were from Roche Diagnostics Gmbl (Mannheim, Germany).

### 2.2. Cell Culture

Cancer cell lines: A549 lung carcinoma, MCF7 breast adenocarcinoma, T24 bladder transitional cell carcinoma, PANC-1 pancreas epithelioid carcinoma and U-2 OS osteosarcoma were from American Type Tissue Collection (ATCC, USA) and were authenticated using short tandem repeat (STR) analysis. Primary ovarian cancer cells were isolated from ascites fluid, obtained after signing informed consent. All cells were propagated in RPMI1640 supplemented with 10% fetal bovine serum, 2 mM L-glutamine and antibiotics in a 37 °C humidified incubator with 5% CO_2_.

### 2.3. Botanical Extracts

Standardized dried 1:5 water herbal extracts were purchased from BARA (Yokneam, Israel). The dried powder was dissolved in PBS at a concentration of 100 mg/mL, and incubated at 60 °C for 30 min with occasional vortex. The solution was centrifuged at 5000 rpm for 5 min, and the supernatant was filtered through a 0.45 µM Millex PVDF filter (Millipore, Carrigtwohil, Ireland). Solubility was estimated by cryophilization and weighting of the pellet and was estimated to be about 50%. For convenience, the final stock concentration was designated at 100 mg/mL (*w*/*v* concentration of dried powder in PBS).

### 2.4. Classification-Guided Two-Botanical Combination Screening

The botanicals were classified into three groups according to their anti-cancer effect and its dependence on ROS induction (Table 1; for detailed classification please refer to [25]): non-toxic botanicals (NT. Botanicals that had no inhibitory effect on cancer cells growth at 2 mg/mL concentration), toxic non-ROS (TNR, botanicals that had anti-cancer effect in vitro, but this effect was not affected by ROS neutralization by pyruvate) and toxic ROS-inducing (TR, botanicals, which anti-cancer effect in vitro could be neutralized by pyruvate addition). Different approaches were used for each kind of combinations:(1)Combinations of NT botanicals (which previously were found to be non-toxic at 2 mg/mL concentration on a panel of several cancer cell lines) were screened by a single dose of 2 mg/mL of 1:1 mix on a panel of cancer cell lines as indicated in the figures. In the case of synergy, the inhibition of growth should appear, and we should be able to detect it on the viability graph.(2)Combinations of TR botanicals, theoretically working through similar mechanism, were tested by comparing dose-effect curves of single botanicals and 1:1 mix. The combination was considered synergic if the mix had stronger effect than each of the individual herbs at similar doses, antagonistic if it had weaker effect than each of the individual herbs and additive if its effect was between the individual botanicals.(3)Combinations of toxic (TNR or TR) botanicals with NT botanicals were tested by comparing the effect of a single 1 mg/mL dose of the toxic botanical to the same treatment with addition of 1 mg/mL of NT botanical. In case of synergy, the effect of toxic botanical should be improved, in case of antagonism—compromised, in case of no effect (or additivity)—unchanged.(4)For combinations of TR and TNR botanicals three parameters were considered: (a) the influence of TNR on the effect of TR in TNR-insensitive cells (similarly to toxic:non-toxic combinations, were the TNR botanical was considered as “non-toxic”); (b) Combined dose-effect curves versus single herbs curves (as for TR botanicals); (c) Combinational index (CI), calculated using CompuSyn program (ComboSyn Inc, Paramus, NJ, USA, 2005. www.combosyn.com, accessed on 12 November 2022, [26]).(5)Combinations of TNR botanicals were evaluated by comparing dose-effect curves of single botanicals to 1:1 mix and calculating CI.

### 2.5. SRB Viability Assay

Cells were plated 3000/w over 96 well plates and allowed to attach and grow overnight. Treatments were added for indicated times. SRB viability test was performed as described [27] in the following way: the cells were fixed for 1 h with 10% trichloroacetic acid (*v*/*v* in RPMI1640), washed trice with double distilled water, dried and stained for 30 min with 0.057% SRB (*w*/*v* in 1% acetic acid). After staining, the plates were washed trice with 1% acetic acid, dried and 200 µL 10 mM Tris was added to each well to solubilize SRB. Absorbance was measured at 570 nm using Power Wave X 340-I ELISA reader (Biotek Instruments, Winooski, VT, USA). Each experiment was repeated at least three times in triplicates.

### 2.6. FACS Analysis

Cells were plated on a 10^6^/10 cm plate, and treated on the following day. The cells were harvested by trypsinization at indicated times, fixed with 70% ice-cold ethanol, stained with PI/RNAse A/Tryton X-100 mix for 40 min, and used for cell cycle and apoptosis analyses. Cell sorting was performed on a BD FACS Calibur flow cytometer (BD Biosciences, San Jose, CA, USA). Single-cell population was selected on FL-2A/FL-2W dotplot and analyzed on an FL-2A linear scale using a WinMDI 2.9 program (Purdue University Cytometry Laboratories, West Lafayette, IN, USA).

### 2.7. Immunoblotting

Cells were plated at a density of 10^6^/10 cm plate and treated as indicated in the legends on the following day. Protein was extracted using RIPA (150 mM NaCl, 1% NP-40, 0.5% deoxycholic acid, 0.1% SDS, 0.5 M Tris pH 8), supplemented with a complete mini protease inhibitor cocktail and phosphatase inhibitor cocktails 2 and 3. Protein concentration was determined with Pierce BCA protein assay kit (Thermo Scientific, Rockford, IL, USA). Samples (30–100 µg) were resolved on 10–12% SDS PAGE, transferred to Protran BA-83 0.2 µM nitrocellulose membrane (Whatman, Piscataway, NJ, USA), blocked with 5% BSA and immunoblotted with appropriate antibodies. The membrane was then washed thrice with tris buffered saline, with Tween (TBST), incubated with corresponding HRP-conjugated secondary antibodies, probed with EZ-ECL enhanced chemiluminescence detection kit and then exposed to Fuji Super RX film (Fujifilm, Tokyo, Japan).

### 2.8. Statistical Methods

The mean ± standard deviations were calculated for each treatment, performed in triplicates, unless indicated otherwise. *p*-values were calculated using Student’s *t*-test. All data were collected and analyzed using Microsoft Excel 2007 (version 12.0, Microsoft, Washington, DC, USA).

### 2.9. Combinational Index

Combinational Index was calculated using CompuSyn program (ComboSyn Inc, Paramus, NJ, USA, 2005. www.combosyn.com, accessed on 12 November 2022, [26]).

## 3. Results

### 3.1. Screening of Two-Botanical Combinations

We performed classification-guided screening of two-botanical combinations for several botanicals (please refer to Table 1 for full names and acronyms), previously classified by us, according to their anti-cancer effect, into three groups: TR, TNR and NT.

The screening was performed as described in the “Methods” section on A549, MCF7, T24 and PANC-1 human cancer cell lines according to their sensitivity to the tested herbs, obtained in the previous work [25]. MCF7 cells are AME-sensitive; PANC-1 and T24 are VHI-sensitive, A549 cells are insensitive to both. Thus, to show the effect of VHI combinations with NT botanicals, we selected one insensitive (A549) and one sensitive (PANC-1) cell lines. Likewise, for TR combinations, we selected T24 and PANC-1 cells, which are mildly sensitive to ROS, and can provide more stable results. In addition, some combinations of toxic botanicals were tested on primary ovarian cancer cells. Figure 1 shows several examples of this classification-guided screening:

(A) 1:1 combinations of eight NT botanicals, screened at a single dose on four cancer cell lines, showing no effect;

(B) Influence of eight NT botanicals on the effect of VHI (TNR) in VHI-insensitive A549 cells and VHI-sensitive PANC-1 cells, screened by addition of NT botanicals to a fixed dose of VHI and showing no significant effect. Both TNR botanicals, AME and VHI, were screened in combinations with all NT botanicals on A549, MCF7, T24 and PANC-1 cell lines, neither combination showed clear synergy or antagonism, only selected results (with one sensitive and one insensitive cell lines) are shown;

(C) 1:1 combination of TR botanicals (PLW and SSU), tested by comparing dose-effect curves of the mix and single botanicals, displaying expected additive effect (the dose-effect curve of 1:1 mix is between the curves of single botanicals);

(D–E) 1:1 combinations of TR and TNR botanicals, tested in rising concentrations on different cells.

As shown in Figure 1D, AME (TNR) blocked the effect of PVU (TR) in AME-insensitive primary ovarian cancer cells, suggesting interference between these two herbs. Similar interference was also observed in VHI-sensitive PANC-1 cells, where the combination of VHI (TNR) and SSU (TR) had lower effect than had each of the single botanicals, and combinational index indicated strong interference.

### 3.2. Synergic Growth Inhibition and Cancer Cells Death Induction by AME:VHI Extract

The synergic effect of AME:VHI (AV) extract was initially discovered in primary ovarian cancer cells, and lately observed during two-herb combinations screening in several cancer cell lines, as described in the previous section. Human A549 (NSCLC), U-2 OS (osteosarcoma), MCF7 (breast adenocarcinoma), T24 (bladder transitional cell carcinoma) and PANC-1 (pancreas epithelioid carcinoma) cell lines were treated with rising concentrations (0.125–2.0 mg/mL) of AME, VHI or 1:1 mix (AV). As shown in Figure 2A, AV inhibited growth of all five cell lines at much lower concentrations than either AME or VHI alone. Combinational index, calculated using CompuSyn program, confirmed that the combined effect was either synergic (U2-OS, PANC-1) or strongly synergic (A549, MCF7 and T-24). Strong synergic effect was achieved with primary ovarian cancer cells, insensitive to both botanicals (Figure 2B). To further investigate the mechanism of the synergy, we selected three cell lines showing strong synergic inhibition and lower sensitivity to single botanicals: A549 (almost insensitive to both botanicals), MCF7 (AME-sensitive, VHI-insensitive) and T-24 (VHI-sensitive, AME-insensitive). This was done to filter as much as possible the “noise” from single botanicals effect and get a better chance to find markers relevant for the synergy. Several experiments were also performed on VHI-sensitive, AME-insensitive PANC-1 cells.

Since viability experiments at a single time point provide only relative viability of control and treated populations, and cannot distinguish between growth inhibition and cell death, we performed cell cycle analysis of AV-treated cells. Cell death through apoptosis induction was shown by elevated sub-G1 population in cells treated with AV for 72 h (Figure 2C, cell cycle analysis by FACS) and confirmed by PARP-1 cleavage (Figure 2D, Western blot). Interestingly, cell cycle analysis showed elevation of G2/M phase in all cells, and of S phase in A549, T24 and PANC-1 (Figure 2E); these results were further supported by accumulation of cyclin A, a marker of cell cycle progression and especially G2 phase, in AV-treated cells (Figure 2D). Further, we evaluated time-dependent effect of AV and compared it to the effect of single botanicals by treating A549, MCF7, T24 and PANC-1 cells with 2 mg/mL of AME, VHI or AV for 264 h (the treatments were replaced by fresh ones every 72 h). As shown in Figure 3, even cells sensitive to AME or VHI alone overcame the inhibiting effect of single botanicals over time. On the contrary, AV treatment induced time-dependent cell death in three cell lines (A549, MCF7 and PANC-1), reducing cell population far below the pre-treatment values. T24 cells entered growth arrest after 72 h of treatment.

Taken together, these results demonstrate that AME and VHI synergically induce growth arrest and death of cancer cells in vitro.

### 3.3. AV’s Effect on Proliferation Pathways and Cell Cycle Progression

To address the molecular mechanism underlying AV’s effect, we first analyzed the key enzymes in major proliferation and survival pathways, commonly activated in cancer cells, PI3K/Akt/mTOR and MAP kinase pathways [28,29]. Surprisingly, AV induced hyper phosphorylation of Akt (on both Ser473 and Thr308) in all three cell lines and ERK1/2 hyper phosphorylation in A549 and MCF7 cells at 24 h treatment (Figure 4A top). In T24 cells ERK1/2 hyper phosphorylation was observed after 72 h of treatment (Figure 4B), consistent with delayed effect of AV on these cells (Figure 3). Interestingly, AV treatment had much lower effect, if any, on the phosphorylation of MEK1/2, an upstream activator of ERK1/2. Simultaneously, 24 h treatment reduced the level of cell cycle progression inhibitors p21 and p27 in all cells (Figure 4A top). Cell cycle analysis showed that 24 h AV treatment elevated the proportion of cells in G2/M phase in all three cell lines, and in S phase in A549 and T24 (Figure 4A bottom), similarly to the results seen upon 72 h treatment (Figure 2E). This is consistent with the growth curves (Figure 3) which showed that all cells continued to proliferate during the first 24 h of treatment. Similar effect of AV on cell cycle distribution was observed under serum starvation conditions (Figure 4C top). AV abrogated the starvation-induced reduction in the G2/M phase in all three cell lines, and reduction in the S phase in A549 and T-24. Coherently, starvation did not rescue the cells from AV-induced death (Figure 4C bottom), but rather had an additive effect with AV treatment.

Altogether, these results suggest that AV’s mechanism of action may involve aberration of cell cycle progression signals and/or checkpoints.

### 3.4. IGF1-R Inhibitor AEW-541 Attenuates AV’s Effect

To test the role of Akt and ERK1/2 hyper activation in AV’s effect on cancer cells, we utilized inhibitors of Akt and ERK1/2 upstream activators, namely, a MEK1/2 inhibitor, Selumetinib (MEK1/2 is ERK1/2 direct upstream activator), a PI3K inhibitor, ZSTK474 (PI3K is an upstream activator of Akt), and IGF1R inhibitor AEW-541 (IGF1R activates both Akt and ERK1/2 pathways). As shown in Figure 5B, inhibition of MEK1/2 and PI3K successfully blocked AV-induced hyper phosphorylation of ERK1/2 and Akt, respectively, but failed to rescue the cells (Figure 5A). On the contrary, AEW-541 drastically attenuated the effect of AV on the viability of the cells (Figure 5C). Moreover, AEW-541 also attenuated the VHI effect on VHI-sensitive T24 and PANC-1 cells, but not the AME effect on AME-sensitive MCF7 cells, suggesting that rescue of the cells by AEW-541 may be due to the blockage of the VHI effect on the cells. Consistently, AEW-541 treatment blocked Akt and ERK1/2 hyper phosphorylation and restored p21 and 27 levels in AV-treated cells (Figure 5D).

## 4. Discussion

We present here a new method to screen potential anti-cancer botanical combinations, based on their classification according to ROS involvement in their mechanism of action. Since ROS involvement in anti-cancer activity of botanicals is very common, as shown previously [25,30], our classification and screening method can be a very effective tool in future finding of new potentially useful combinations. The main idea of this screening method is at the first step to distinguish between toxic (to cancer cells) and non-toxic botanicals by simple viability assay after treatment with the botanicals of choice; then to repeat the assay with potentially toxic botanicals using a ROS-neutralizing agent pyruvate [8], thereby distinguishing between ROS-dependent and ROS-independent botanicals. At the third stage, the costumed screening presented here have the following advantages:(1)It allows to avoid multiple dose points required for CI calculations when at least one of the botanicals is non-toxic, and to perform the synergy screen by single-dose treatment;(2)It allows similar one-dose screening for TNR and TR botanicals pairs in TNR-insensitive cells;(3)Suggests that ROS inducing botanicals (TR) probably have an additive effect, as expected from substances with similar mechanism of action [25]. Altogether, our screening method allows a simple, organized and effective way for screening of potentially synergic combinations and suggests including the analysis of ROS involvement as a necessary step, since according to our previous finding, the majority of potentially anti-cancer botanicals work through this mechanism.

This approach led us to the discovery of a strong synergic combination of two botanicals—*Astragalus membranaceus* and *Vaccaria hispanica*, AV. Both botanicals were previously shown to inhibit cancer cells proliferation [4,13,14,15,16,17,20,31,32]. However, the combined effect of *A. membranaceus* and *V. hispanica* is investigated here for the first time. We previously showed that both botanicals inhibit cancer cells through ROS-independent mechanisms [25]. Moreover, since they selectively inhibit different sets of cells [25], they probably work through different mechanisms.

Dose-effect and time-effect curves showed that the AV combination is not only much more effective than each of the botanicals alone, but is also effective against cells insensitive to either of them. There are two possible explanations for this phenomenon: either one of the botanicals accelerates\improves the action of the other, thereby turning it effective against previously insensitive cells, or the combination works through a different mechanism. Additional studies are required to answer this question.

Cancer cells are often characterized by hyper activation of proliferation and survival pathways. Consequently, investigators of new drugs usually expect to find inhibition of these pathways [33]. However, here we observed a very different situation: Akt and ERK1/2 pathways were hyper activated by AV treatment, while the cell cycle control proteins p21 and p27 were strongly down regulated, but despite this apparent stimulation of key proteins in proliferation and survival pathways, the AV treatment led to the inhibition of proliferation and induction of apoptosis. Moreover, the cells accumulated in G2/M phase of the cell cycle, as shown by cyclin A accumulation and FACS analysis. Altogether, these results suggest that AV treatment interferes with cell cycle, possibly pushing cells to proliferate beyond their ability. The fact that AV treatment abrogated starvation-induced slow-down of proliferation, further supports this hypothesis. Pushing cancer cells to proliferation by loosening checkpoint proteins in order to increase replicative stress has been proposed previously as a potential anti-cancer strategy [34]. However, there are currently no available drugs working through such mechanism.

Interestingly, prevention of AV-induced Akt or ERK1/2 hyper activation by PI3K and MEK1/2 inhibitors, respectively, did not rescue the cells. However, IGF1R inhibitor AEW-541 rescued all AV-treated cells and VHI-treated PANC-1 and T24 (VHI-sensitive) cells, suggesting that the VHI anti-cancer mechanism may be part of AV’s effect. As expected, AEW-541 prevented hyper activation of Akt and ERK1/2 and partially abrogated p21 and p27 down regulation. However, additional studies are needed to address the role of these proteins in the AV mechanism of action.

The major limitation of this study is mainly descriptive and initial nature of the investigation of the mechanism of AV action. The second limitation is the lack of the studies of the effect of each individual botanical and its constituents (which can vary depending on the climate change, geographic region, harvesting process and so on) on the biological effect of AV and the analyzed pathways. Isolating the relevant compounds may enhance the observed synergistic effect and promote the investigation of exact mechanism of action by reducing the influence of irrelevant compounds. In addition, we have to validate our findings in an in vivo system. These issues will be addressed in further studies.

## 5. Conclusions

Altogether, our results show that the AV combination, discovered through our classification-based screening method, has strong synergic anti-cancer effects, probably involving interference with the cell cycle and loosening cell cycle checkpoints. Additional studies are required to validate the effect of the AV combination on a wide range of cancer cell types, in order to better understand its mechanism of action and its potential as an anti-cancer therapy.

## Figures and Tables

**Figure 1 cancers-14-05833-f001:**
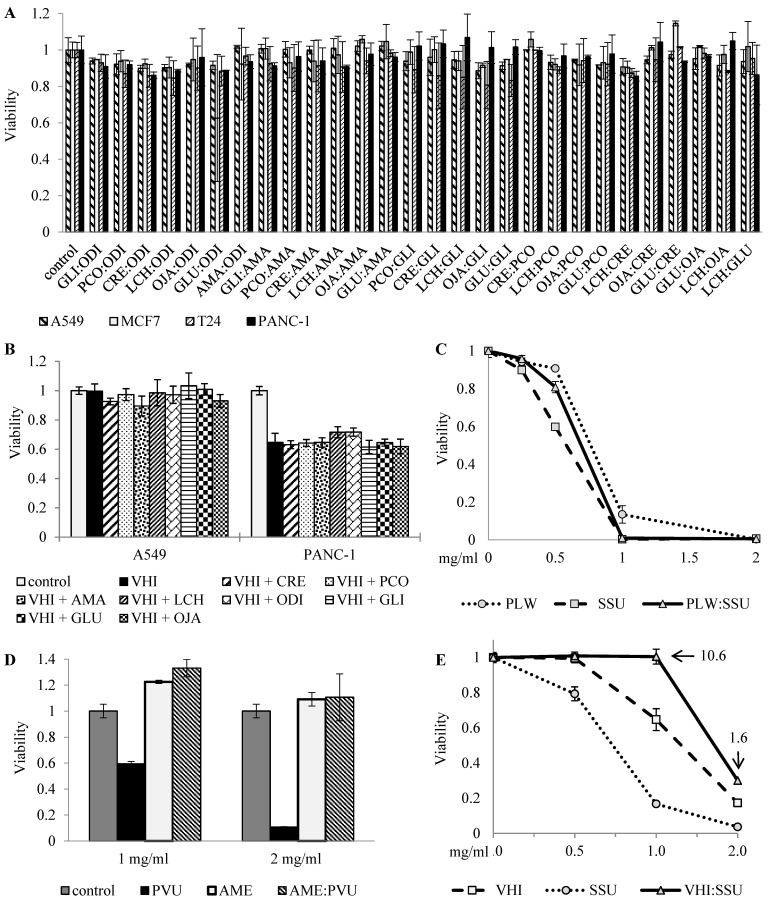
Classification-guided screening two-herb combinations for anti-cancer effect. (**A**) A549, MCF7, T24 and PANC-1 cells were treated for 72 h with 2 mg/mL 1:1 mixes of indicated non-toxic botanicals (average of two independent experiments); (**B**) A549 and PANC-1 cells were treated for 72 h with 1 mg/mL of VHI either alone or with 1 mg/mL of indicated non-toxic botanicals; (**C**) T24 cells treated for 72 h with rising concentrations of ROS-inducing botanicals PLW and SSU or their 1:1 mix; (**D**) Primary ovarian cancer cells treated for 72 h with indicated concentrations of toxic botanicals AME (non-ROS), PVU (ROS) or 1:1 mix; (**E**) PANC-1 cells treated for 72 h with rising concentrations of toxic botanicals VHI (non-ROS), SSU (ROS) or 1:1 mix (the numbers represent combinational index at arrow-indicated points (10.6 at 1 mg/mL and 1.6 at 2 mg/mL).

**Figure 2 cancers-14-05833-f002:**
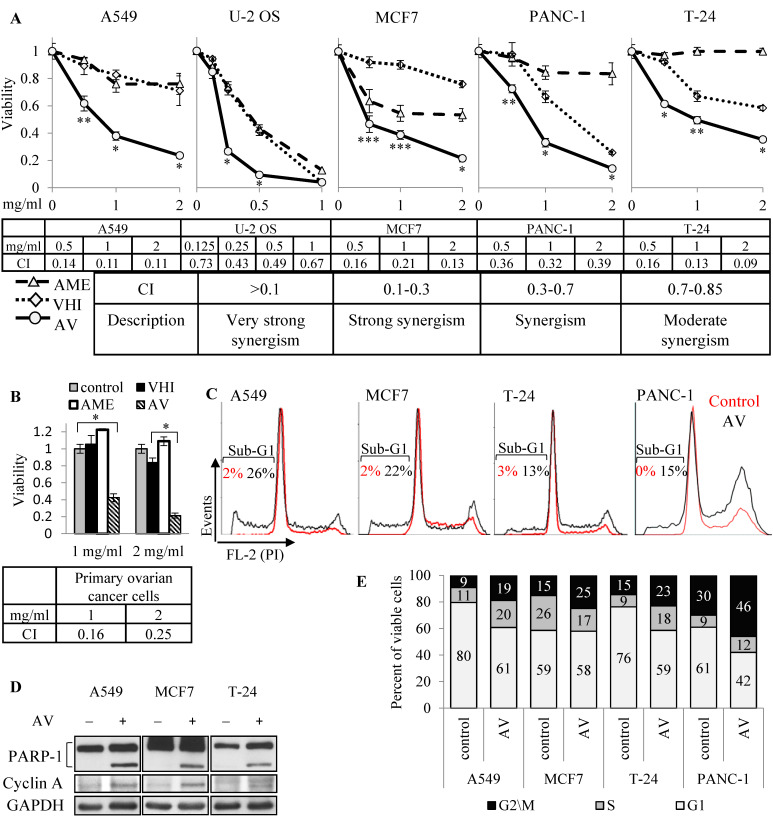
Synergic AME and VHI toxicity towards cancer cells. (**A**) Viability of cancer cell lines upon 72 h treatment by rising concentrations of AME, VHI or 1:1 mix (AV). Combinational Index (CI, table below the graphs), calculated using CompuSyn program; (**B**) Same as in (**A**) for primary ovarian cancer cells; (**C**) Cell cycle analysis of cells treated with 2 mg/mL of AV for 72 h, showing sub-G1 (apoptotic) population (the graphs are normalized to similar G1 peak in control and treatment samples). (**D**) Western blot of cells treated with 2 mg/mL of AV for 72 h showing apoptotic PARP-1 cleavage and cyclin A accumulation; (**E**) Same as (**C**), showing percent of cells in G1, S and G2/M phases, calculated from non-apoptotic population.*—*p*-value < 0.001, **—*p*-value < 0.01, ***—*p*-value < 0.05, as compared between AV point and the closest relevant value of other compared groups (control or single botanicals). CI – combinational index; G1, S, G2/M – phases of cell cycle; FL-2 (PI) – fluorescence channel – 2 (propidium iodide).

**Figure 3 cancers-14-05833-f003:**
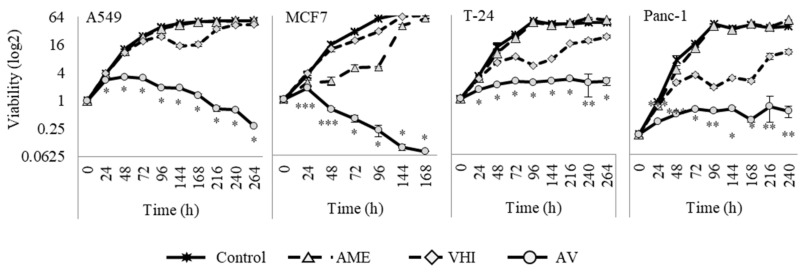
Time-dependent effect of AME, VHI and AV on cancer cells. The cells were treated with 2 mg/mL of AME, VHI or AV and checked for viability at indicated times after the beginning of the treatment. The medium was replaced every 72 h with similar control/treatment-containing medium. The experiment for each cell line was continued either for 264 h after the beginning of the treatment or until the viability of AV-treated cells decreased below 10% of 0 time viability. All values were normalized to the viability of “0” time (the time of treatment addition). Please note that the viability scale is logarithmic (log2). *—*p*-value < 0.001, **—*p*-value < 0.01, ***—*p*-value < 0.05, as compared between AV point and the closest relevant value of other compared groups (control or single botanicals).

**Figure 4 cancers-14-05833-f004:**
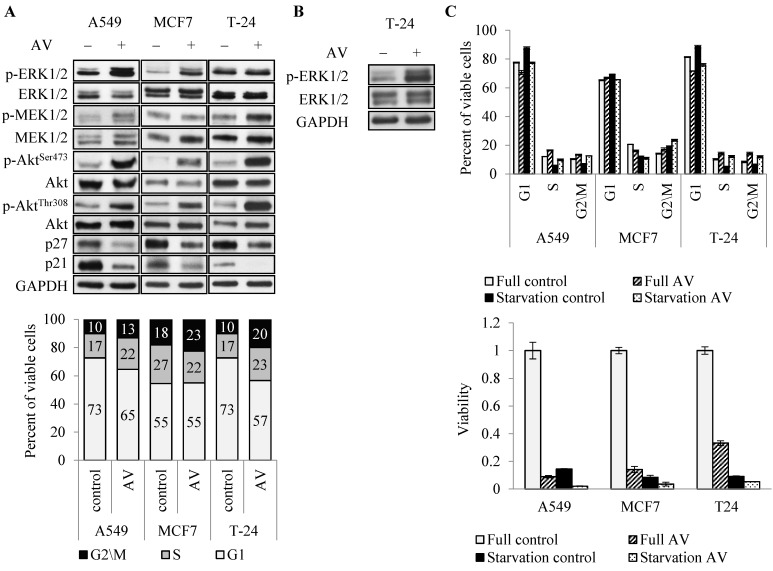
AV’s effect on proliferation pathways and cell cycle progression. (**A**) Cells were treated for 24 h with 2 mg/mL AV and either analyzed on Western blot for protein expression (upper panel) or subjected to cell cycle analysis by FACS (bottom panel, percent of viable cells in G0/G1, S and G2/M phases); (**B**) Western blot of T24 cells, treated with 2 mg/mL AV for 72 h; (**C**) AV (2 mg/mL) effect on cells in serum starvation conditions. Upper panel—distribution of cell cycle after 24 h treatment; bottom panel—viability after 72 h.

**Figure 5 cancers-14-05833-f005:**
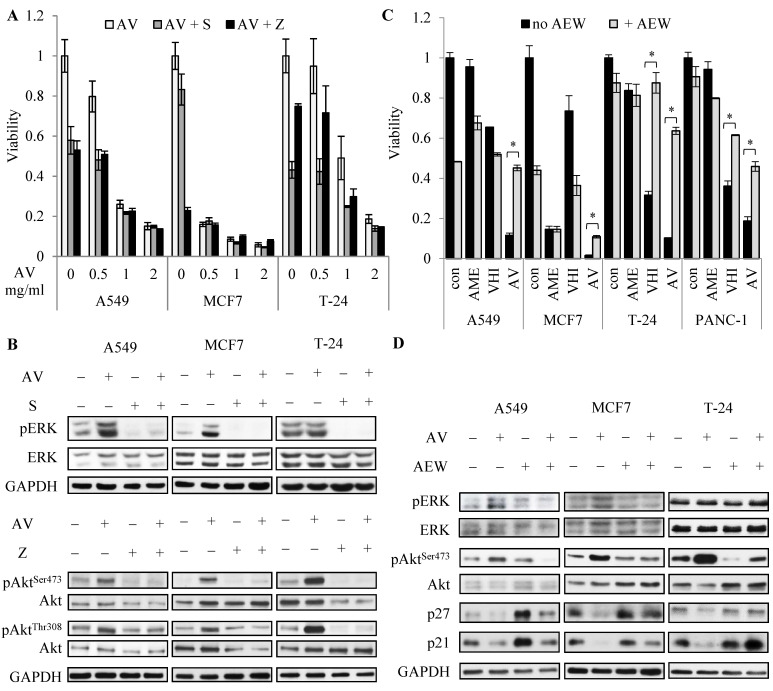
Influence of MEK1/2, PI3K and IGF1-R inhibitors on AV’s effect. (**A**) Viability of cells treated with AV for 72 h in the presence or absence of 2 mM Selumetinib (“S”, MEK1/2 inhibitor) or 1 mM ZSTK-740 (“Z”, PI3K inhibitor). (**B**) WB of cells treated with 2 mg/mL AV with or without 2 mM Selumetinib or 1 mM ZSTK-740 for 24 h. (**C**) Viability of cells treated for 72 h with 2 mg/mL of AME/VHI/AV with or without 1 µM AEW541. (**D**) Western blot of cells treated for 24 h similarly to (**C**). *—*p*-value < 0.001. S – Selumetinib; Z - ZSTK-740; AEW - AEW541.

**Table 1 cancers-14-05833-t001:** Botanical names and abbreviations.

Acronym	Latin Name	ROS and Toxicity Classification
AMA	*Atractylodes macrocephala*	NT
AME	*Astragalus membranaceus*	TNR
CRE	*Citrus reticulata*	NT
GLI	*Glehnia littoralis*	NT
GLU	*Ganoderma lucidum*	NT
LCH	*Lycium chinense*	NT
OJA	*Ophiopogon japonicus*	NT
ODI	*Oldenlandia diffusa*	NT
PCO	*Poria cocos*	NT
PLW	*Paeonia lactiflora white*	TR
PVU	*Prunella vulgaris*	TR
SSU	*Spatholobus suberectus*	TR
VHI	*Vaccaria hispanica*	TNR

## Data Availability

The data presented in this study are available on request from the corresponding author.

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
