# Peer review of "Strong Synergic Growth Inhibition and Death Induction of Cancer Cells by Astragalus membranaceus and Vaccaria hispanica Extract"

_cancers, 2022, doi:10.3390/cancers14235833_

Round 1

Reviewer 1 Report

1. Name of the plants, In vitro and In vivo should be in italics throughout the manuscript.

2. Authors have used the term "Standardized dried herbal extracts". What are the standards? Herbal extracts? Are they water extracts or organic solvent extracts? Please mention. 

3. 600C should be 60 degrees Celsius.

4. In table 1 authors should include anti-cancer effect (TR, TNR and NT) of each plant.

5. Authors have used 13 plant combinations. What about the results of other combinations besides AME:VHI (AV). Do they not show any effect?

------------

Author Response

Reviewer 1:

  1. Name of the plants, In vitro and In vivo should be in italics throughout the manuscript.

 Answer: corrected throughout the manuscript, as requested by the reviewer (can be viewed by “track changes”).

  1. Authors have used the term "Standardized dried herbal extracts". What are the standards? Herbal extracts? Are they water extracts or organic solvent extracts? Please mention. 

Answer: we added “1:5 water” (page 3, line 100).

  1. 600C should be 60 degrees Celsius.

Answer: corrected (line 102 now), as well as the same typo error on line 156. Thank you for the remark.

  1. In table 1 authors should include anti-cancer effect (TR, TNR and NT) of each plant.

 Answer:

a)  We added the classifications of the plants to Table 1.

b) Line 109: we added reference to Table 1.

  1. Authors have used 13 plant combinations. What about the results of other combinations besides AME:VHI (AV). Do they not show any effect?

Answer: the screening examples presented in Figure 1 are shown to illustrate the screening approach rather than to provide the screening results of all the combinations.

a) All NT botanical combinations are shown (no synergy/antagonism is detected).

b) Both TNR botanicals (AME and VHI) were screened in combinations with all NT botanicals in all four cell lines, neither combination showed synergy or antagonism (only some results are shown). We added description of all the results on page five, starting at line 193.

c) Combinations with TR botanicals are difficult to evaluate due to sharp decline in dose-viability curves ones ROS death threshold is reached. We show some combinations, in which results we are completely confident. However, none of other tested combinations showed clear synergic effect. 

English corrections and spelling:

  • Page 1, line 29: “a” was deleted.
  • Page 5, line 205: “had” was added.
  • Page 9, line 279: repeated “the” was deleted.
  • Page 9, line 281: extra space was deleted.
  • Page 12, line 374: “totally” was replaced by “very”
  • Page 13, line 409: “a” was deleted.

Reviewer 2 Report

Authors developed a method to screen potential anti-cancer botanical combinations and provide evidences of a strong synergic combination of two botanicals, Astragalus membranaceus and Vaccaria hispanica.

The screening approach is interesting and can provide useful information. However, the use of different cell lines to test the various combination of botanicals makes difficult to compare the results. Considering the different effect of some botanicals on single cell lines, this approach is quite confusing. Probably some preliminary data are available from previous works and could be resumed at the beginning of the “Results” section. A better description of the screening approach, including the reason for the choice of specific cell lines to test the effect of different and selected combinations of botanicals (NT vs NT, TR vs TR and  TR vs TNR)  should be included  (i. e. why only VHI was selected as TNR to test vs NT, why the choice of  use/show only selected combinations ). Additionally, in table 1 the classification of the different botanicals according to their anti-cancer effect (TR, TNR and NT) is not reported (readers must find it in the previous work or deduct it from the text).

Indeed, the results obtained strongly suggest the synergic effect of the selected combination.  In this case, the second part of the experimental work has been conducted on selected cell lines and results were more correctly compared. The use of specific antibodies and inhibitors provide enough information on the mechanism of action and the cellular pathways involved, even if additional studies are needed.

Other minor points

Figure 5: why Authors used different concentration of AV in panel A. In all other panel a fixed concentration was used.  Additionally, in panel B the exact AV concentration used is not reported.

Line 100: 600C should be 60°C

Line 147: 106 should be 10^6

Line 212: WHE should be WHI

Author Response

1) The screening approach is interesting and can provide useful information. However, the use of different cell lines to test the various combination of botanicals makes difficult to compare the results. Considering the different effect of some botanicals on single cell lines, this approach is quite confusing. Probably some preliminary data are available from previous works and could be resumed at the beginning of the “Results” section. A better description of the screening approach, including the reason for the choice of specific cell lines to test the effect of different and selected combinations of botanicals (NT vs NT, TR vs TR and  TR vs TNR)  should be included  (i. e. why only VHI was selected as TNR to test vs NT, why the choice of  use/show only selected combinations).

Answer: The screening examples presented in Figure 1 are shown to illustrate the screening approach rather than to provide the screening results of all the combinations. We added an explanation on our selection of specific cell lines to show the combinations effect on page five, starting at line 182. Additionally:

a) All NT botanical combinations with all four cell lines are shown (no synergy/antagonism is detected).

b) Both TNR botanicals (AME and VHI) were screened in combinations with all NT botanicals in all four cell lines, neither combination showed synergy or antagonism (only some results are shown). We added description of all the results on page five, starting at line 193.

c) Combinations with TR botanicals are difficult to evaluate due to sharp decline in dose-viability curves ones ROS death threshold is reached. We show some combinations (on mildly ROS-sensitive cells), in which results we are completely confident. However, none of other tested combinations showed clear synergic effect. 

2) Additionally, in table 1 the classification of the different botanicals according to their anti-cancer effect (TR, TNR and NT) is not reported (readers must find it in the previous work or deduct it from the text).

      Answer:

a) We added the classifications of the plants to Table 1.

b) Line 109: we added reference to Table 1.

Other minor points

1) Figure 5: why Authors used different concentration of AV in panel A. In all other panel a fixed concentration was used.  Additionally, in panel B the exact AV concentration used is not reported.

 Answer:

a) We used a range of AV concentrations in panel A since it is a dose-effect curve, like in Figure 1. We think that showing a range of concentrations is important, and show it wherever is possible. However, we can remove it upon request.

b) We added the exact AV concentration and the concentrations of the inhibitors to the Figure 5 legend, panel B.

2) Line 100: 600C should be 60°C

Answer: corrected (line 102 now), thank you for the remark.

3) Line 147: 106 should be 10^6

Answer: corrected (line 148 now), as well as the same typo error on line 156. Thank you for the remark.

4) Line 212: WHE should be WHI

Answer: we corrected the typo error on line 212 (line 226 now) from VHE to VHI, thank you for the remark.

Author Response

Cancers Manuscript ID.: cancers-2010330 Title: Strong synergic growth inhibition and death induction of cancer cells by Astragalus membranaceus and Vaccaria hispanica extract The topic is really interesting. However, I believe author should go through the comments and modify manuscript accordingly so as to get possible chance of its acceptance Major Comments The main components of A. membranaceus are Astragalus polysaccharide (APS), flavonoids compounds, saponins compounds, alkaloids, etc. APS is the most important natural active component in A. membranaceus, and possesses multiple pharmacological properties.

1) We need to understand the pharmacological actions of APS. Kindly provide a detailed view about pharmacological action of APS

2) Regulation effect of APS on immunity (APS promotes humoral immune response by regulating the functional activity of natural killer and natural killer T cells. How they promote???

3) Inhibiting the Metastasis of Tumor Cells APS can significantly inhibit the metastasis of C33A cells in cervical cancer, which may be achieved by reducing the levels of matrix metalloproteinase 2 (MMP2)??

4) Nevertheless, further clarification of its mechanisms involved in reducing blood lipids, antagonizing fibrosis, bacteriostasis, radiation protection, and antiviral activities is warranted.

5) At present, injection with APS has been used to assist in radiotherapy and chemotherapy, and play a synergistic role in reducing toxicity. APS is a natural complex compound; thus, the greatest current challenge in APS research is to extract its specific components and identify their precise targets. Accurate detection of the targets of the nine pharmacological actions of APS and demonstration of the remarkable effects of the multi-target integration of traditional Chinese medicine will be more instructive for the clinical use of APS.

Answer to points 1-5: our article is not a review on A. membranaceus, and thus in the discussion we had to concentrate on most relevant points to our results. We studied only the anticancer mechanism of the Astragalus –Vaccaria combination in vitro, and we do know yet which components of the botanicals provide the observed synergy. However, we added description on the Astragalus and APS, regarding the points mentioned by the reviewer (page 12, starting on line 357, and references to some excellent reviews (18-19). If any additional reference should be added, please let us know.

6) Role of Crude polysaccharides and the aqueous extract from the seeds of V. Hispanica in treating benign prostatic hyperplasia.

Answer: we added the relevant sentence on page 13, line 356 and reference 17.

7) Mode of administration of V hispanica??

Answer: In our manuscript, we used only cell cultures, thus the extracts were directly added to the cell culture media (described in materials and methods section). If you meant something else please let us know and we will correct the manuscript accordingly.

  • Page 1, line 29: “a” was deleted.
  • Page 5, line 205: “had” was added.
  • Page 9, line 279: repeated “the” was deleted.
  • Page 9, line 281: extra space was deleted.
  • Page 12, line 374: “totally” was replaced by “very”
  • Page 13, line 409: “a” was deleted.

Round 2

Reviewer 2 Report

Authors addressed all quetions.

Author Response

Dear reviewer,

Thank you for your review.

Reviewer 3 Report

Authors have not clearly responded the comments provided in (comments 1 till 6). we are familiar about manuscript originality whether it is research article or review article.

These are some additional information author is being asked to add.

Author Response

Dear reviewer,

We addressed now your comments 1- 5 in the introduction (page 2, lines 52-82);

comment 6 is addressed in the introduction (page 2, lines 83 - 91).

We also added a sentence addressing the need for detailed compounds study regarding our results in the limitation section of the Discussion (page 13, lines 423 - 425).

Please note that several references were added and as a result reference's numbering was shifted (the changes are highlighted by "track changes" function).

Thank you for your review, please let us know if you would like us to add additional information. 

Kind regards.

Round 3

Reviewer 3 Report

Manuscript may be accepted for publication